

# A morphological and molecular study of *Hydrodynastes gigas* (Serpentes, Dipsadidae), a widespread species from South America

Priscila S. Carvalho[1,2], Hussam Zaher[3], Nelson J. da Silva Jr[4] and Diego J. Santana[1]

[1] Instituto de Biociências, Universidade Federal de Mato Grosso do Sul, Campo Grande, Mato Grosso do Sul, Brazil
[2] Instituto de Biociências, Letras e Ciências Exatas, Universidade Estadual Paulista, São José do Rio preto, São Paulo, Brazil
[3] Museu de Zoologia da Universidade de São Paulo, São Paulo, São Paulo, Brazil
[4] Escola de Ciências Médicas, Farmacêuticas e Biomédicas, Pontifícia Universidade Católica de Goiás, Goiânia, Goiás, Brazil

Corresponding author
Priscila S. Carvalho,
pricarvalho.bio@gmail.com

## ABSTRACT

**Background**. Studies with integrative approaches (based on different lines of evidence) are fundamental for understanding the diversity of organisms. Different data sources can improve the understanding of the taxonomy and evolution of snakes. We used this integrative approach to verify the taxonomic status of *Hydrodynastes gigas* (Duméril, Bibron & Duméril, 1854), given its wide distribution throughout South America, including the validity of the recently described *Hydrodynastes melanogigas Franco, Fernandes & Bentim, 2007*.

**Methods**. We performed a phylogenetic analysis of Bayesian Inference with mtDNA 16S and Cytb, and nuDNA Cmos and NT3 concatenated (1,902 bp). In addition, we performed traditional morphometric analyses, meristic, hemipenis morphology and coloration pattern of *H. gigas* and *H. melanogigas*.

**Results**. According to molecular and morphological characters, *H. gigas* is widely distributed throughout South America. We found no evidence to support that *H. gigas* and *H. melanogigas* species are distinct lineages, therefore, *H. melanogigas* is a junior synonym of *H. gigas*. Thus, the melanic pattern of *H. melanogigas* is the result of a polymorphism of *H. gigas*. Melanic populations of *H. gigas* can be found in the Tocantins-Araguaia basin.

## INTRODUCTION

Species are considered lineages with distinct evolutionary histories (*De Queiroz, 2007*). Taxonomic studies are traditionally based on morphological descriptors to delimit species (e.g., *Franco et al., 2017*; *França et al., 2018*; *Meneses-Pelayo & Passos, 2019*). However, in many cases, species are difficult to delimit due to the limited number of

morphological differences or the absence of them, preventing the recognition of valid cryptic species (*Bickford et al., 2007*). Morphology alone might result in more than one name being assigned to individuals belonging to the same evolutionary lineage (i.e., species) (*Passos & Prudente, 2012*; *Passos, Martins & Pinto-Coelho, 2016*; *Mângia et al., 2020*). Many species are described based solely on morphological patterns, which could merely reflect interpopulational variation, instead of evidence of lineage separation (e.g., *Brusquetti et al., 2014*; *Mângia et al., 2020*).

Currently, species delimitation must be based on the integration of more than one type of data set (e.g., DNA sequences, morphology, behavior, pheromone), which helps improve taxonomic understanding (*Dayrat, 2005*; *Padial & De La Riva, 2010*; *Padial et al., 2010*; *Pante, Schoelinck & Puillandre, 2014*). An integrative approach contributes to taxonomic, phylogenetic and phylogeographic studies (*Pante, Schoelinck & Puillandre, 2014*), being thus useful in delimiting species and sorting possible interspecific variations and lineages with similar morphologies. Ultimately, integrative approaches are essential for testing taxonomic schemes and correcting nomenclatural inconsistencies (e.g., *Recoder et al., 2014*; *Ruane et al., 2018*; *Mângia et al., 2020*).

Among the morphological characteristics adopted in taxonomic studies that usually result in inaccurate nomenclatural decisions are coloration pattern variations (e.g., *Atractus* spp.: *Passos, Martins & Pinto-Coelho, 2016*; *Apostolepis* spp.: *Entiauspe-Neto et al., 2019*). Animal coloration has several adaptive functions, including thermoregulation, signaling and protection (*Briolat et al., 2019*). Variation in coloration patterns (polychromatism), often associated with ontogenetic dimorphism (e.g., *Corallus* spp.: *Henderson, 1997*; *Henderson, Passos & Feitosa, 2009*; and *Drymoluber* spp.: *Costa, Moura & Feio, 2013*) or with chromatic anomalies such as leukism, albinism and melanism (the latter caused by the increase of epidermal pigments known as melanin; *Kettlewell, 1973*) can lead to erroneous nomenclatural decisions. Although there are some exclusively melanistic species (e.g., some species of the Pseudoboini tribe), melanism can also be the result of intraspecific polymorphism (e.g., *Bernardo et al., 2012*). The adaptive value of melanism may be related to predation, protection or thermoregulation (*Andrén & Nilson, 1981*; *Forsman & Ås, 1987*; *Capula, Luiselli & Monney, 1995*; *Briolat et al., 2019*).

*Hydrodynastes* Fitzinger, 1843 is a genus of large semiaquatic snakes, which currently contains three species: *Hydrodynastes bicinctus* (Hermann, 1804) distributed in Colombia, Venezuela, French Guiana, Guyana, Suriname and Brazil (*Murta-Fonseca, Franco & Fernandes, 2015*); *Hydrodynastes gigas* (*Duméril, Bibron & Duméril, 1854*) distributed in French Guiana, Bolivia, Paraguay, Argentina and Brazil (*Giraudo & Scrocchi, 2002*; *Pereira-Filho & Montingelli, 2006*; *Wallach, Williams & Boundy, 2014*; *Nogueira et al., 2019*); and *Hydrodynastes melanogigas* (*Franco, Fernandes & Bentim, 2007*) recorded only in the Tocantins-Araguaia Basin in the states of Tocantins, Mato Grosso and Maranhão, Brazil (*Silva Jr et al., 2012*; *Santos Jr et al., 2017*). *Hydrodynastes gigas* (*Duméril, Bibron & Duméril, 1854*) has a wide distribution throughout Brazil, and has been recorded in the states of Amapá, Amazonas, Pará, Rondônia, Roraima, Tocantins, Maranhão, Piauí, Rio Grande do Norte, Mato Grosso, Mato Grosso do Sul, Minas Gerais, São Paulo, Paraná, Rio Grande do Sul (*Nogueira et al., 2019*).
Given the wide distribution of *Hydrodynastes gigas* in South America, a level of intraspecific variation throughout its populations is to be expected, with some potential to represent still undescribed cryptic species. Indeed, *Franco, Fernandes & Bentim (2007)* considered one of these populations as a distinct species, describing *H. melanogigas* mainly through its differential color pattern and pointed out its similarity with *H. gigas* on meristic and hemipenial characteristics. Therefore, the distinction between *H. melanogigas* and *H. gigas* rests mainly on its melanistic color pattern and on its inferred allopatric distribution with *H. gigas*. The present study aims to evaluate the taxonomic validity of *Hydrodynastes gigas* and *Hydrodynastes melanogigas* using an integrative taxonomic approach inferred by molecular and morphological data.

## MATERIALS & METHODS

We evaluated the taxonomic status of *Hydrodynastes gigas* and *Hydrodynastes melanogigas* by sequencing two mitochondrial and two nuclear genes for 32 individuals belonging to the two species. We further analyzed the external morphology of 186 specimens of *H. gigas* and *H. melanogigas*.

### Molecular analysis

We extracted the DNA from muscle, liver or scale of 27 samples of *Hydrodynastes gigas*, five of *H. melanogigas* and 12 of *H. bicinctus*. Samples of *Hydrodynastes bicinctus* were added to our analysis in order to provide a complete species representation for the genus. We used the phenol-chloroform extraction protocol (*Sambrook, Fritsch & Maniatis, 1989*) (Fig. 1A). We amplified the partial sequences of the mitochondrial 16S ribosomal (mtDNA) genes (16S rRNA, 326 pb) (*Palumbi et al., 2002*), Cytochrome b (Cytb, 618 pb) (*Pook, Wüster & Thorpe, 2000*), the nuclear genes (nuDNA) Oocyte maturation factor Mos (Cmos, 478 pb) (*Lawson et al., 2005*) and Neurotrophin-3 (NT3, 480 pb) (*Townsend et al., 2008*) using the standard Polymerase Chain Reaction (PCR) technique as described by (*Pook, Wüster & Thorpe, 2000*) and *Moraes-Da-Silva et al. (2019)*. We visually checked all nucleotide sequences and aligned the concatenated genes (1,902 pb) using the Muscle algorithm (*Edgar, 2004*) in the Geneious v.9.0.5 program (https://www.geneious.com). We used the species *Pseudoboa nigra* and *Xenopholis scalaris* as outgroup of *Hydrodynastes* (*Vidal, Dewynter & Gower, 2010*; *Zaher et al., 2009*), and rooted the tree in *Xenopholis*. We used the sequences available in GenBank and deposited those generated in this study into the same database (Table 1).

We used PartitionFinder 2 to identify partitioning schemes and the most appropriate nucleotide replacement models (*Lanfear et al., 2016*). According to our concatenated alignment, we found five partitions evaluated by BIC (Table 2). For phylogenetic analysis, we used the Bayesian inference implemented in MrBayes v3.2.6 (*Ronquist & Huelsenbeck, 2003*) using the substitution models generated by PartitionFinder. We ran two independent runs of four Markov chains for 20 million generations sampling every 5,000 generations and discarding 25% as burn-in. We evaluated the stability of the analysis in Tracer v1.6, ensuring that all ESS values were above 200 (*Rambaut et al., 2014*). We calculated the divergence between sequences (p-distance) in Mega v10.0.5 (*Tamura et al., 2013*).
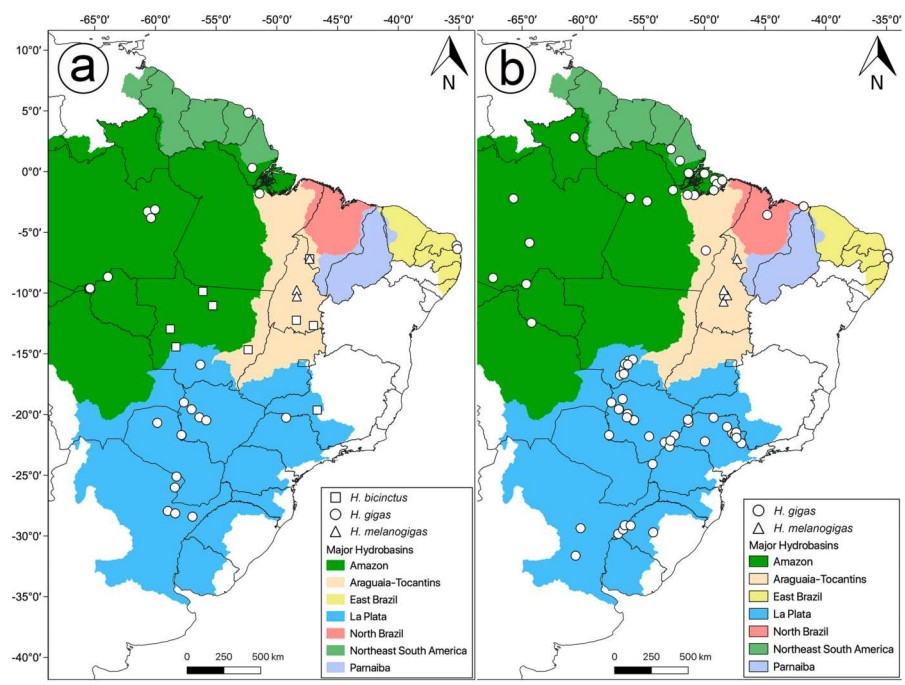

**Figure 1** **Distribution of analyzed data in this study.** Sample localities for (A) molecular and (B) morphology of *Hydrodynastes* analyzed in this study.

## Morphological analysis

We examined 144 specimens of *Hydrodynastes gigas* and 42 specimens of *H. melanogigas* (Fig. 1B). Museum acronyms follow *Sabaj-Pérez (2014)*, except for Coleção Herpetológica da Universidade Federal da Paraíba (CHUFPB), João Pessoa, PB; Coleção Zoológica Delta do Parnaíba (CZDP), Parnaíba, PI; Universidade Luterana do Brasil (MZCEULP), Palmas, TO. The specimens examined are listed in the Appendix S1.

*Franco, Fernandes & Bentim (2007)* described *Hydrodynastes melanogigas* based on 17 specimens collected in the municipalities of Palmas (type locality), Porto Nacional, and Lajeado, which are all located in the state of Tocantins, Brazil. Unfortunately, most of the type series was lost in the 2010 fire that occurred at the Instituto Butantan. Currently, only three individuals remain from the type series: two at the proper Institute in São Paulo (paratypes IBSP 65978 and IBSP 66387) and one at the Museu Nacional do Rio de Janeiro (paratype MNRJ 15101). From the remaining type-series, we analyzed all the remaining individuals.

We examined 14 meristic characters (Table 3) and eight morphometric ones (Table 4), in addition to the coloration pattern and morphology of the hemipenis. Sex was determined by the presence or absence of hemipenes through a ventral incision at the base of the tail. We measured individuals with an electronic caliper (0.01 mm) and a flexible ruler (1 mm), on their right side whenever possible. In order to test morphometric differences between *H. gigas* and *H. melanogigas*, we conducted a principal component analysis (PCA) and took the first two principal components of the ordination to create a MANOVA. We

**Table 1 Voucher information and GenBank numbers.** Specimens used for the molecular analyses, including GenBank numbers for mitochondrial 16S and Cytb, nuclear Cmos and NT3 sequences. * data not available in the original references.

| Species | Voucher | Locality | Genbank Accession number | | | | Reference |
|---|---|---|---|---|---|---|---|
| | | | 16S | Cytb | Cmos | NT3 | |
| *Hydrodynastes bicinctus* | CHUNB52057 | Brazil, Maranhão, Carolina | MT192271 | MT224977 | MT328069 | MT328103 | This study |
| *Hydrodynastes bicinctus* | CHUNB47129 | Brazil, Mato Grosso, Alta Floresta | MT192270 | MT224976 | – | – | This study |
| *Hydrodynastes bicinctus* | CHUNB63637 | Brazil, Mato Grosso, Nova Xavantina | MT192272 | MT224978 | MT328070 | MT328104 | This study |
| *Hydrodynastes bicinctus* | MZUSP17831 | Brazil, Mato Grosso, Itaúba | MT192274 | MT224980 | MT328072 | MT328105 | This study |
| *Hydrodynastes bicinctus* | UFMT7550 | Brazil, Mato Grosso, Tangará da Serra | MT192276 | MT224981 | MT328073 | MT328107 | This study |
| *Hydrodynastes bicinctus* | UFMT7551 | Brazil, Mato Grosso, Tangará da Serra | MT192277 | MT224982 | MT328074 | MT328108 | This study |
| *Hydrodynastes bicinctus* | 1817 (MZUSP) | Brazil, Mato Grosso, Sapezal | MT192267 | MT224972 | MT328065 | MT328100 | This study |
| *Hydrodynastes bicinctus* | MZUSP20580 | Brazil, Rondônia, Porto Velho, Mutum | MT192275 | – | – | MT328106 | This study |
| *Hydrodynastes bicinctus* | CHUNB38982 | Brazil, Tocantins, Arraias | MT192268 | MT224973 | MT328066 | – | This study |
| *Hydrodynastes bicinctus* | CHUNB40618 | Brazil, Tocantins, Mateiros | – | MT224974 | MT328067 | MT328101 | This study |
| *Hydrodynastes bicinctus* | CHUNB40619 | Brazil, Tocantins, Mateiros | MT192269 | MT224975 | MT328068 | MT328102 | This study |
| *Hydrodynastes bicinctus* | MZUSP15560 | Brazil, Tocantins, UHE Peixe Angical | MT192273 | MT224979 | MT328071 | – | This study |
| *Hydrodynastes gigas* | INALI6573 | Argentina, Chaco, San Fernando | MT192283 | MT224989 | – | MT328114 | This study |
| *Hydrodynastes gigas* | INALI4779 | Argentina, Corrientes, General San Martín | MT192282 | MT224988 | – | MT328113 | This study |
| *Hydrodynastes gigas* | INALI6867 | Argentina, Corrientes, Mburucuya | MT192285 | MT224991 | – | MT328116 | This study |
| *Hydrodynastes gigas* | LGE7992 | Argentina, Formosa, Pirané | MT192287 | MT224995 | – | MT328119 | This study |
| *Hydrodynastes gigas* | INALI6731 | Argentina, Formosa, Pilcomayo | MT192284 | MT224990 | – | MT328115 | This study |
| *Hydrodynastes gigas* | MZUSP11704 | Brazil, Amapá | MT192289 | MT225000 | – | MT328124 | This study |
| *Hydrodynastes gigas* | INPA-HT5513 | Brazil, Amazonas, Careiro | MT215328 | MT224994 | MT328081 | MT328118 | This study |
| *Hydrodynastes gigas* | INPA-HT190 | Brazil, Amazonas, Manaus | MT215327 | MT224992 | MT328079 | MT328117 | This study |
| *Hydrodynastes gigas* | MTR19444 | Brazil, Amazonas, Manacapuru | MT215333 | MT224999 | MT328086 | MT328123 | This study |
| *Hydrodynastes gigas* | CHUNB65028 | Brazil, Mato Grosso, Nossa Senhora do Livramento | MT192281 | MT224986 | MT328077 | MT328111 | This study |

| Species | Voucher | Locality | Genbank Accession number | | | | Reference |
|---------|---------|----------|------|------|------|-----|-----------|
| | | | 16S | Cytb | Cmos | NT3 | |
| *Hydrodynastes gigas* | MAP-T3894 | Brazil, Mato Grosso do Sul, Porto Murtinho | MT215330 | MT224997 | MT328083 | MT328121 | This study |
| *Hydrodynastes gigas* | MAP4050 | Brazil, Mato Grosso do Sul, Porto Murtinho | MT215329 | MT224996 | MT328082 | MT328120 | This study |
| *Hydrodynastes gigas* | ZUFMS-REP2392 | Brazil, Mato Grosso do Sul, Anastácio | MT192296 | MT225007 | MT328092 | MT328130 | This study |
| *Hydrodynastes gigas* | ZUFMS-REP 2393 | Brazil, Mato Grosso do Sul, Corumbá | MT192297 | MT225008 | MT328093 | MT328131 | This study |
| *Hydrodynastes gigas* | ZUFMS-REP 2395 | Brazil, Mato Grosso do Sul, Miranda | MT192298 | MT225009 | MT328094 | MT328132 | This study |
| *Hydrodynastes gigas* | ZUFMS-REP 2476 | Brazil, Mato Grosso do Sul, Corumbá | MT192299 | MT225010 | MT328095 | MT328133 | This study |
| *Hydrodynastes gigas* | ZUFMS-REP 2389 | Brazil, Minas Gerais, Fronteira | MT192295 | MT225006 | MT328091 | MT328129 | This study |
| *Hydrodynastes gigas* | MPEG21864 | Brazil, Pará, Melgaço | MT192288 | MT224998 | MT328084 | MT328122 | This study |
| *Hydrodynastes gigas* | AAGARDA8745 | Brazil, Rio Grande do Norte, Nísia Floresta | MT192278 | MT224983 | MT328075 | MT328109 | This study |
| *Hydrodynastes gigas* | AAGARDA12357 | Brazil, Rio Grande do Norte, Canguaretama | MT192279 | MT224984 | – | – | This study |
| *Hydrodynastes gigas* | INPA-HT5427 | Brazil, Rondônia, Porto Velho, Teotônio | MT192286 | MT224993 | MT328080 | – | This study |
| *Hydrodynastes gigas* | MZUSP18572 | Brazil, Rondônia, Porto Velho, Mutum | MT192290 | MT225001 | MT328087 | MT328125 | This study |
| *Hydrodynastes gigas* | MZUSP18573 | Brazil, Rondônia, Porto Velho, Abunã | MT192291 | MT225002 | MT328088 | MT328126 | This study |
| *Hydrodynastes gigas* | MZUSP19710 | Brazil, Rondônia, Porto Velho, Abunã | MT192292 | MT225003 | MT328089 | MT328127 | This study |
| *Hydrodynastes gigas* | MZUSP20449 | Brazil, Rondônia, Porto Velho, Abunã | MT192293 | MT225004 | MT328090 | MT328128 | This study |
| *Hydrodynastes gigas* | AF2382 | French Guiana, Matoury | MT192280 | MT224985 | MT328076 | MT328110 | This study |
| *Hydrodynastes gigas* | PINV1580254 | Paraguay, Alto Paraguay | MT192294 | MT225005 | – | MT424769 | This study |
| *Hydrodynastes melanogigas* | MPEG24383 | Brazil, Maranhão, Carolina | MT215331 | MT225012 | – | MT328097 | This study |
| *Hydrodynastes melanogigas* | MPEG24384 | Brazil, Maranhão, Carolina | MT215332 | MT225013 | MT328085 | MT328098 | This study |
| *Hydrodynastes melanogigas* | MZUSP19557 | Brazil, Maranhão, Carolina | MT215334 | MT225014 | – | MT328099 | This study |
| *Hydrodynastes melanogigas* | IBSP65144 | Brazil, Tocantins, Lajeado | – | MT224987 | MT328078 | MT328112 | This study |
| *Hydrodynastes melanogigas* | UFMS-REP3446 | Brazil, Tocantins, Palmas | MT215335 | MT225011 | MT328096 | MT328134 | This study |
| *Pseudoboa nigra* | MZUSP13278 | * | GQ457764 | JQ598948 | GQ457885 | – | *Zaher et al. (2009)* and *Grazziotin et al. (2012)* |

**Table 1** (*continued*)

| Species | Voucher | Locality | Genbank Accession number | | | | Reference |
|---------|---------|----------|------|------|------|-----|-----------|
| | | | **16S** | **Cytb** | **Cmos** | **NT3** | |
| *Xenopholis scalaris* | KU222204 | * | JQ598915 | GQ895897 | JQ599002 | – | *Pyron et al. (2011)* and *Grazziotin et al. (2012)* |

**Table 2  PartitionFinder 2 model of nucleotide substitution.** Best-fitting partitioning scheme model of nucleotide substitution for 16S, Cytb, Cmos and NT3 genes.

| Partitioning scheme | Model |
|---------------------|-------|
| Cytb1, 16S | GTR+G |
| Cytb2, Cmos2 | HKY+I |
| Cytb3 | HKY+G |
| NT32, Cmos3, Cmos1 | JC |
| NT31, NT33 | K80+I+G |

ran this analysis with adult males and females separately, and performed the analysis in R software (*R Core Team, 2014*) using the package Vegan (*Oksanen et al., 2007*). We followed the terminology of *Dowling (1951)* for counting the ventral scales, and *Peters (1964)* and *Vanzolini, Ramos-Costa & Vitt (1980)* for pholidosis. We surveyed the geographic coordinates of the data catalogs of zoological collections using Google Earth software.

## Hemipenial morphology

We prepared a hemipenis from a topotype of *Hydrodynastes melanogigas* and 19 from *H. gigas* from different localities in the Amazon, East Brazil and La Plata hydrobasins (Appendix S1). Whenever possible, we prepared the hemipenes on the right side according to the technique originally described by *Manzani & Abe (1988)*, as modified by *Pesantes (1994)*, *Zaher (1999)*, and *Zaher & Prudente (2003)*. We stained the external calcareous structures with alizarin red, as suggested by *Nunes et al. (2012)*, for a better visualization of microstructures in the surface of the organ. Terminology follows *Dowling & Savage (1960)*, *Zaher (1999)* and *Myers & Cadle (2003)*.

## RESULTS

### Molecular approach

We recovered the genus *Hydrodynastes* as monophyletic, and the topology of the concatenated gene tree showed two strongly supported clades with posterior probability ($pp = 1.00$). Our concatenated dataset tree grouped *Hydrodynastes melanogigas* within *H. gigas* (Fig. 2). The uncorrected p-distance for both the mtDNA 16S and Cytb showed low genetic differences (0.01% and 0.2%, respectively) between the lineages of *H. gigas* and *H. melanogigas.* However, the genetic differences between *H. gigas* and *H. bicinctus* were 0.43% for 16S and 13% for Cytb (Table 5). Intraspecific variation in *H. gigas* was 0.0% to 0.04% for 16S and 0.0% to 0.17% for Cytb, while in *H. bicinctus* it was 0.0% for 16S and 0.0% to 0.23% for Cytb (Supplementary Material).

**Table 3 Meristic characters in *Hydrodynastes gigas* and *Hydrodynastes melanogigas*.**

| Variables | *Hydrodynastes gigas* | | | *Hydrodynastes melanogigas* | |
| --- | --- | --- | --- | --- | --- |
| | Male | Female | Undetermined[**] | Male | Female |
| SLl | 8 ($n = 64$); 9 ($n = 1$) | 8 ($n = 61$); 9 ($n = 5$) | 8 ($n = 6$) | 8 ($n = 22$) | 8 ($n = 17$) |
| ILr | 9 ($n = 5$); 10 ($n = 37$); 11 ($n = 21$); 12 ($n = 1$) | 9 ($n = 2$); 10 ($n = 29$); 11 ($n = 31$); 12 ($n = 5$) | 10 ($n = 3$); 11 ($n = 2$) | 8 ($n = 1$); 9 ($n = 2$); 10 ($n = 17$); 11 ($n = 2$) | 9 ($n = 1$); 10 ($n = 14$); 11 ($n = 2$) |
| Ill | 9 ($n = 4$); 10 ($n = 32$); 11 ($n = 26$) | 9 ($n = 3$); 10 ($n = 24$); 11 ($n = 35$); 12 ($n = 4$) | 10 ($n = 3$); 11 ($n = 3$) | 9 ($n = 4$); 10 ($n = 16$); 11 ($n = 2$) | 10 ($n = 10$); 11 ($n = 6$); 12 ($n = 1$) |
| LO | 1 ($n = 68$) | 1 ($n = 69$) | 1 ($n = 7$) | 1 ($n = 22$) | 1 ($n = 17$) |
| PE | 1 ($n = 66$); 2 ($n = 1$) | 1 ($n = 68$); 2 ($n = 1$) | 1 ($n = 7$) | 1 ($n = 20$); 2 ($n = 2$) | 1 ($n = 16$); 2 ($n = 1$) |
| PO | 2 ($n = 48$); 3 (19) | 1 ($n = 1$); 2 ($n = 64$); 3 ($n = 4$) | 2 ($n = 7$) | 2 ($n = 19$); 3 ($n = 3$) | 2 ($n = 17$) |
| SO | 2 ($n = 3$); 3 ($n = 64$) | 2 ($n = 4$); 3 ($n = 65$) | 3 ($n = 7$) | 3 ($n = 22$) | 3 ($n = 17$) |
| AT | 1 ($n = 5$); 2 (60); 3 ($n = 2$) | 1 ($n = 5$); 2 ($n = 62$); 3 ($n = 2$) | 2 ($n = 5$); 3 ($n = 1$) | 2 ($n = 16$); 3 ($n = 5$); 4 ($n = 1$) | 2 ($n = 17$) |
| PT | 1+2 ($n = 5$); 1+3 ($n = 6$); 2+1 ($n = 1$); 2+2 ($n = 3$); 2+3 ($n = 49$); 2+4 ($n = 2$); 3+3 ($n = 2$) | 2 ($n = 1$); 3 ($n = 1$); 1+2 ($n = 3$); 1+3 ($n = 9$); 2+2 ($n = 7$); 2+3 ($n = 45$); 2+4 ($n = 1$); 3+2 ($n = 1$); 3+3 ($n = 1$) | 2+2 ($n = 2$); 2+3 ($n = 4$); 2+4 ($n = 1$) | 1+3 ($n = 3$); 2+1 ($n = 1$); 2+2 ($n = 2$); 2+3 ($n = 15$) | 1+3 ($n = 3$); 2+2 ($n = 2$); 2+3 ($n = 12$) |
| NA | 2 ($n = 68$) | 2 ($n = 69$) | 2 ($n = 7$) | 2 ($n = 22$) | 2 ($n = 17$) |
| IL+G1 | i-iv ($n = 15$); i-v ($n = 48$); i-vi ($n = 3$) | i-ii ($n = 1$); i-iv ($n = 10$); i-v ($n = 54$); i-vi ($n = 3$) | i-iv ($n = 2$); i-v ($n = 5$) | i-iv ($n = 1$); i-v ($n = 19$); i-vi ($n = 2$) | i-v ($n = 16$); i–iv ($n = 1$) |
| IL+G2 | 0 ($n = 2$); v-vi ($n = 7$); v ($n = 8$); vi ($n = 46$); vii ($n = 3$) | 0 ($n = 3$); v-vi ($n = 5$); v ($n = 4$); vi ($n = 56$); vii ($n = 1$) | v ($n = 2$); vi ($n = 5$) | v ($n = 1$); vi ($n = 19$); vii ($n = 2$) | vi ($n = 15$); v–vi ($n = 1$); vi–vii ($n = 1$) |
| AP | 2 ($n = 67$) | 2 ($n = 69$) | 2 ($n = 4$) | 2 ($n = 22$) | 2 ($n = 17$) |
| AD | 17 ($n = 1$); 18 ($n = 1$); 19 ($n = 59$); 20 ($n = 3$); 21 ($n = 4$) | 18 ($n = 3$); 19 ($n = 54$); 20 ($n = 5$); 21 ($n = 6$) | 19 ($n = 4$) | 19 ($n = 22$) | 19 ($n = 17$) |
| MD | 17 ($n = 2$); 18 ($n = 1$); 19 ($n = 64$) | 17 ($n = 4$); 19 ($n = 65$) | 17 ($n = 1$); 19 ($n = 3$) | 17 ($n = 3$); 19 ($n = 19$) | 18 ($n = 1$); 19 ($n = 16$) |
| PD | 14 ($n = 1$); 15 ($n = 67$) | 14 ($n = 1$); 15 ($n = 67$); 17 ($n = 1$) | 15 ($n = 4$) | 14 ($n = 3$); 15 ($n = 18$); 16 ($n = 1$) | 15 ($n = 17$) |
| PV | 1 ($n = 22$); 2 ($n = 42$); 3 ($n = 4$) | 1 ($n = 18$); 2 ($n = 45$); 3 ($n = 6$) | 1 ($n = 2$); 2 ($n = 1$) | 1 ($n = 10$); 2 ($n = 12$) | 1 ($n = 4$); 2 ($n = 13$) |
| VE | 150–164 | 152–169 | | 154–168 | 168–172 |
| SC[*] | 58(9) –88 | 49(30)–84 | | 71–86 | 70(24) –79 |

**Notes.**

In parenthesis the sampled number (n).

SLr, right supralabials; SLl, left supralabials; ILr, right infralabials; Ill, left infralabials; LO, loreal; PE, preoculars; PO, postoculars; SO, suboculars; AT, anterior temporals; PT, posterior temporals; NA, nasal; IL+G1, infralabials in contact with first pair of genials; IL+G2, infralabials in contact with second pair of genials; AP, apical pits; AD, anterior dorsal scale rows; MD, midbody dorsal scale rows; PD, posterior dorsal scale rows; PV, preventrals; VE, ventral; SC, subcaudal.

[*]75 specimens present autotomized tail.

[**] specimens present only the head or other body part.

**Table 4** Morphometric measurement in *Hydrodynastes gigas* and *Hydrodynastes melanogigas*.

| Variables | *Hydrodynastes gigas* | | *Hydrodynastes melanogigas* | |
| --- | --- | --- | --- | --- |
| | Male | Female | Male | Female |
| SVL | 249–1747 (n = 68) 998 ± 1059,25 | 277–1879 (n = 69) 988.5 ± 512.2 | 468–1548 (n = 22) 1100.4 ± 396.1 | 524–2198 (n = 17) 1048.3 ± 381.4 |
| TL | 84 (25)–580 (n = 68) 302,5 ± 392,44 | 71–543 (n = 69) 276.6 ± 143.6 | 180–548 (n = 22) 351.5 ± 141.5 | 105–532 (n = 17) 355.5 ± 143.6 |
| HL | 20.57–67.94 (n = 63) 44,24 ± 33,49 | 21.65–78.61 (n = 66) 44.5 ± 15.6 | 30.04–55.48 (n = 22) 47.4 ± 7.3 | 32.63–69.21 (n = 17) 46.7 ± 10.3 |
| HW | 7.24–21.54 (n = 67) 10,0 ± 0,17 | 7.19–21.77 (n = 69) 13.9 ± 4.4 | 9.12–17.89 (n = 22) 14.9 ± 2.6 | 9.80–47.25 (n = 17) 15.5 ± 8.5 |
| DN | 2.89–12.93 (n = 68) 10,24 ± 0,71 | 3.51–11.97 (n = 69) 7.6 ± 4.9 | 4.17–9.47 (n = 22) 7.5 ± 1.4 | 4.78–10.68 (n = 17) 7.0 ± 1.6 |
| EN | 3.19–12.55 (n = 68) 9,98 ± 0,18 | 3.70–14.01 (n = 69) 7.3 ± 2.5 | 5.09–9.44 (n = 22) 7.8 ± 1.2 | 4.88–11.58 (n = 17) 7.3 ± 1.8 |
| ED | 4.19–8.55 (n = 68) 6,37 ± 3,08 | 4.12–8.41 (n = 69) 6.1 ± 1.3 | 4.85–7.64 (n = 22) 6.4 ± 0.8 | 5.08–7.95 (n = 17) 6.2 ± 0.8 |
| AC | 7.17–25.63 (n = 61) 10,21 ± 0,33 | 8.03–26.76 (n = 62) 16.3 ± 6.0 | 11.39–21.26 (n = 22) 16.8 ± 2.8 | 9.98–27.64 (n = 17) 15.9 ± 4.4 |

**Notes.**

In parenthesis the sampled number (*n*).

SVL, snout-vent length (from the tip of the snout to the cloaca); TL, tail length; HL, head length (from the tip of the snout to the quadratemandibular articulation); HW, head width (length of the widest part of head); DN, distance between nostrils (maximum distance between the nostrils); EN, distance between eye and nostril; ED, eye diameter; HH, head height (maximum distance between the base of the mandible and the parietal surface.

## Morphological approach

We found overlap in all meristic and morphometric characteristics between *Hydrodynastes gigas* and *H. melanogigas* (Tables 3 and 4). The first principal components from both PCA analysis (males and females) recovered 99% of variation, and through the MANOVA of males ($F = 2.2949$; $p = 0.1109$) and females ($F = 0.3463$; $p = 0.7095$) we found no significant morphometric differences between both species (Fig. 3). In addition, we observed gradient levels of melanism in *H. melanogigas* (Figs. 4A–4L). We examined fully melanic specimens (Fig. 4A) to specimens with clear visible bands along the body (Fig. 4K). We also observed that some *H. gigas* individuals from the Amazon, La Plata and Tocantins-Araguaia basins present darker coloration overlapping the gradient of melanism found in *H. melanogigas* (Figs. 5A–5L). We did not find any morphological characteristics that differentiate the two species.

We did not observe coloration patterns within or between the populations of *Hydrodynastes gigas* (Figs. 6A–6R). We observed ontogenetic variation in the color pattern of all populations analyzed, with no distinction between males and females. Juveniles in the early stages of life have well-defined rounded dark spots all over their backs to the end of their tails; these spots are outlined by a lighter line, while in adults rounded spots may or may not be well defined, and may not present a clear outline (Figs. 7A–7J). Furthermore, we identified two young males of *H. gigas* (CHUNB 22053, Figs. 7I–7J); CHUNB 22068) from the type locality of *H. melanogigas,* as well as 18 more specimens
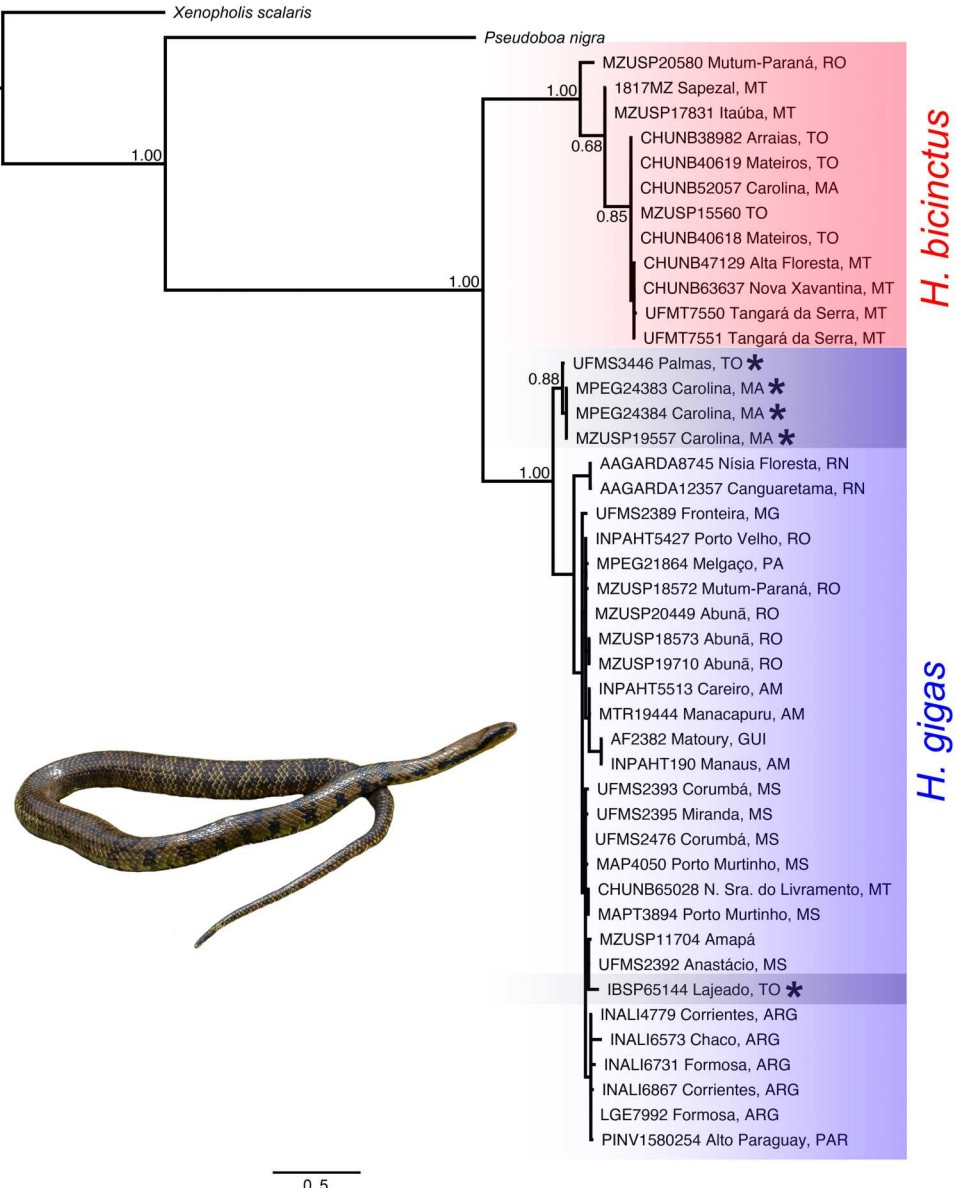

**Figure 2** Concatenated tree (16S, Cytb, C-mos and NT3) of the genus *Hydrodynastes* recovered by Bayesian analysis in MrBayes. Numbers near the nodes correspond to support values indicate by posterior probability (pp). Asterisks indicate samples identified as *H. melanogigas*. Photo credit: Karoline Ceron.

from the Tocantins-Araguaia Basin. We provide more details in the ''variation'' section below.

**Hemipenis morphology** (Figs. 8A–8L): When fully everted and expanded, hemipenes of *H. melanogigas* and *H. gigas* are undistinguishable (Figs. 8K–8L). The hemipenis is deeply bilobed, semicaliculate and semicapitate, with three or four vertical rows of large spines arranged on each side of the body. The body of the hemipenis is covered by spikes on the

**Table 5 Unconrrected *p-distance* of 16S (lower left) and Cytb (upper right) mitochondrial fragment gene for the genus *Hydrodynastes*.** Bold the *p-distance* between *H. gigas* and *H. melanogigas* for the two genes.

|   |   | 1 | 2 | 3 | 4 | 5 |
|---|---|---|---|---|---|---|
| 1 | *Xenopholis scalaris* |  | 0.367 | 0.347 | 0.376 | 0.366 |
| 2 | *Pseudoboa nigra* | 0.084 |  | 0.366 | 0.313 | 0.341 |
| 3 | *Hydrodynastes bicinctus* | 0.126 | 0.118 |  | 0.129 | 0.123 |
| 4 | *Hydrodynastes gigas* | 0.128 | 0.097 | 0.043 |  | **0.020** |
| 5 | *Hydrodynastes melanogigas* | 0.130 | 0.099 | 0.042 | **0.011** |  |

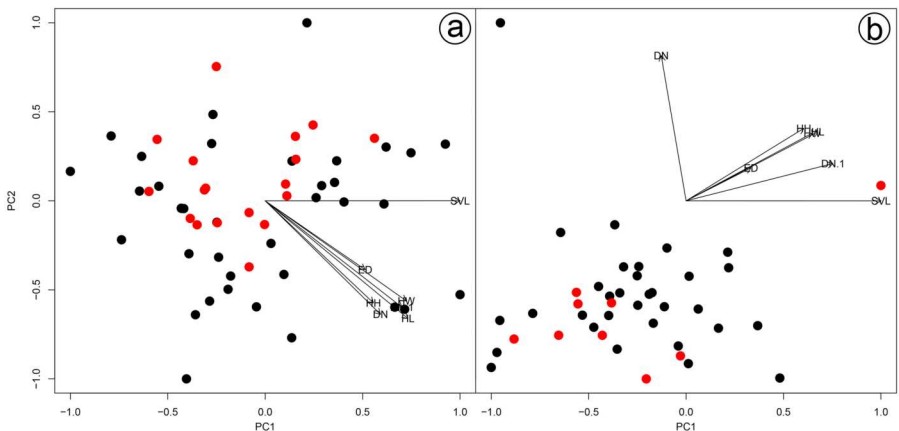

**Figure 3 Results of a Principal Component Analysis (PCA) on the morphometric variables.** Males (A) and females (B) of *Hydrodynastes gigas* and *H. melanogigas*. Black circle corresponds to *H. gigas* and red circle to *H. melanogigas*. SVL, snout-vent length; HL, head length; HW, head width; DN, distance between nostrils; EN, distance between eye and nostril; ED, eye diameter; HH, head height.

sulcate and assulcate faces. The sulcus spermaticus bifurcates in the proximal region of the hemipenis body and each branch extends centrolinearly until it reaches the proximal region of the lobes, in which they follow a centrifugal position that ends at the lateral region of the tip of each. The capitulum, formed by papillate calyces and spikes, extends over most of the surface of the lobes, except in the region of the assulcate face that is occupied by two parallel rows of papillated and conspicuous body calyces that extend to the distal region of the hemipenis body, where they converge on the lobular crest and continues to the middle portion of the hemipenis. We detected low intraspecific variation among *Hydrodynastes gigas* populations. Some hemipenes showed little size variation in lobes and body calyces, varying from slightly visible to conspicuous in the hemipenial body.

## DISCUSSION

*Hydrodynastes gigas* is widely distributed throughout South America, occurring with low genetic variability throughout most of its extension range. Although the genetic structure of widely distributed species can be easily influenced by natural barriers (*Patton, Da Silva*

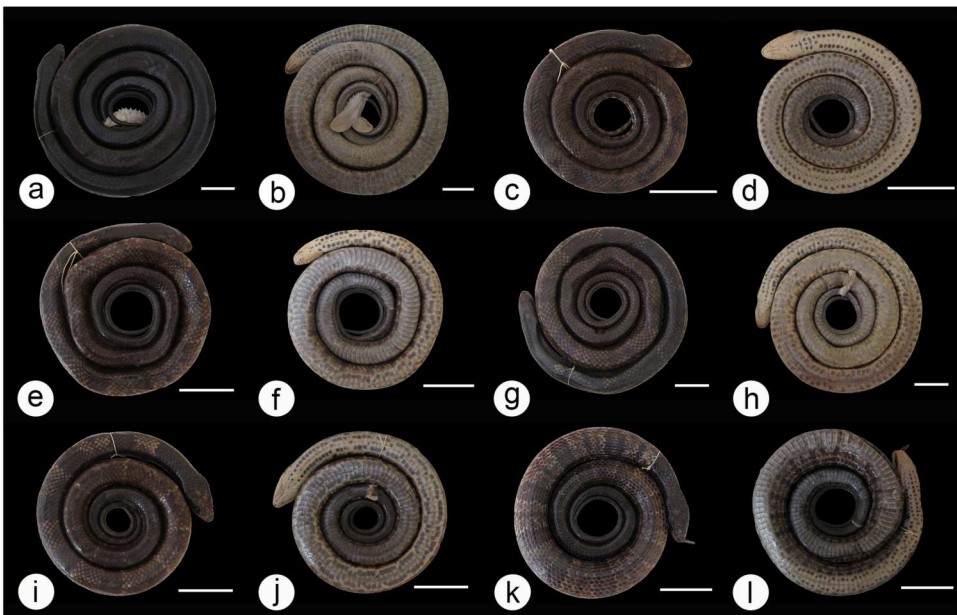

**Figure 4** **Dorsal and ventral view of the melanism gradient in *Hydrodynastes melanogigas*.** MZCEULP 1218 (A, B); MZCEULP 516 (C D,) ; MZCEULP 1046 (E, F); MZCEULP 1273 (G, H); MZCEULP 758 (I, J); MZCEULP 938 (K, L). All specimens from Tocantins-Araguaia basin. Scale of 50 mm.

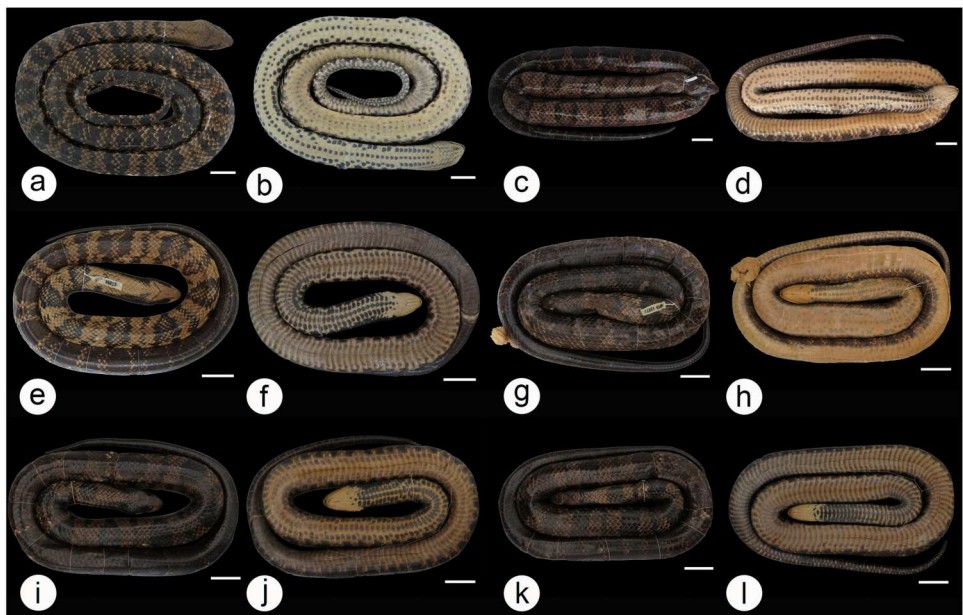

**Figure 5** **Dorsal and ventral view of the coloration gradient in *Hydrodynastes gigas*.** La Plata basin UFMT 9076 (A, B); Amazon basin CHUNB 15159 (C, D); Tocantins-Araguaia basin MPEG 18012 (E, F), MPEG 18071 (G, H), MPEG 18046 (I, J), MPEG 18070 (K, L). Scale of 50 mm.

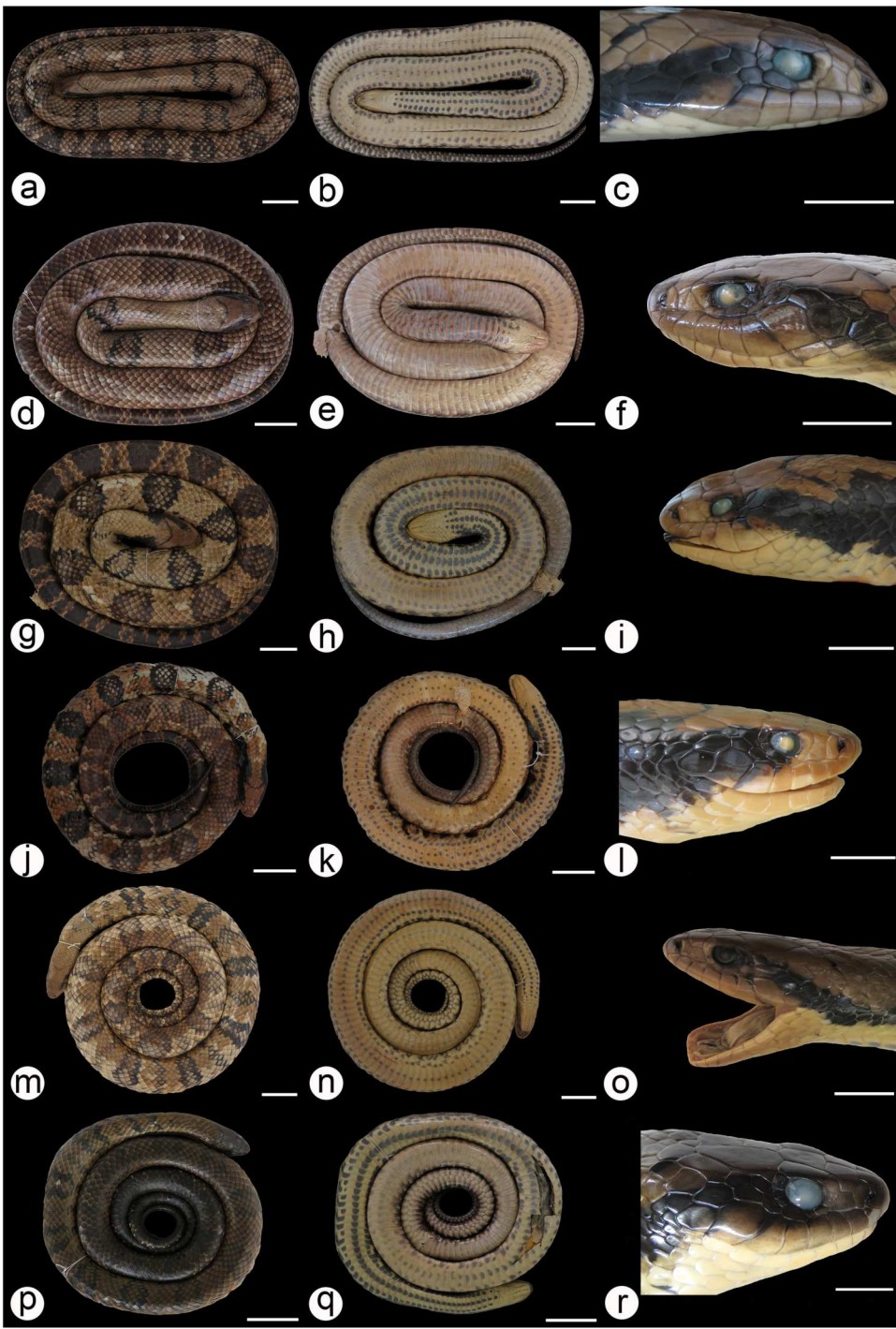

**Figure 6   Dorsal and ventral view of the body and lateral view of the head in *Hydrodynastes gigas*.**
Amazon basin MPEG 22652 (A, B, C), CHUNB 56729 (D, E, F), MPEG 18674 (G, H, I); East Brazil
CHUFPB 4611 (J, K, L); La Plata Basin UFMT 026 (M, N, O), ZUFMS 2389 (P, Q, R). Scale of 50 mm for
body view and scale of 20 mm head view.

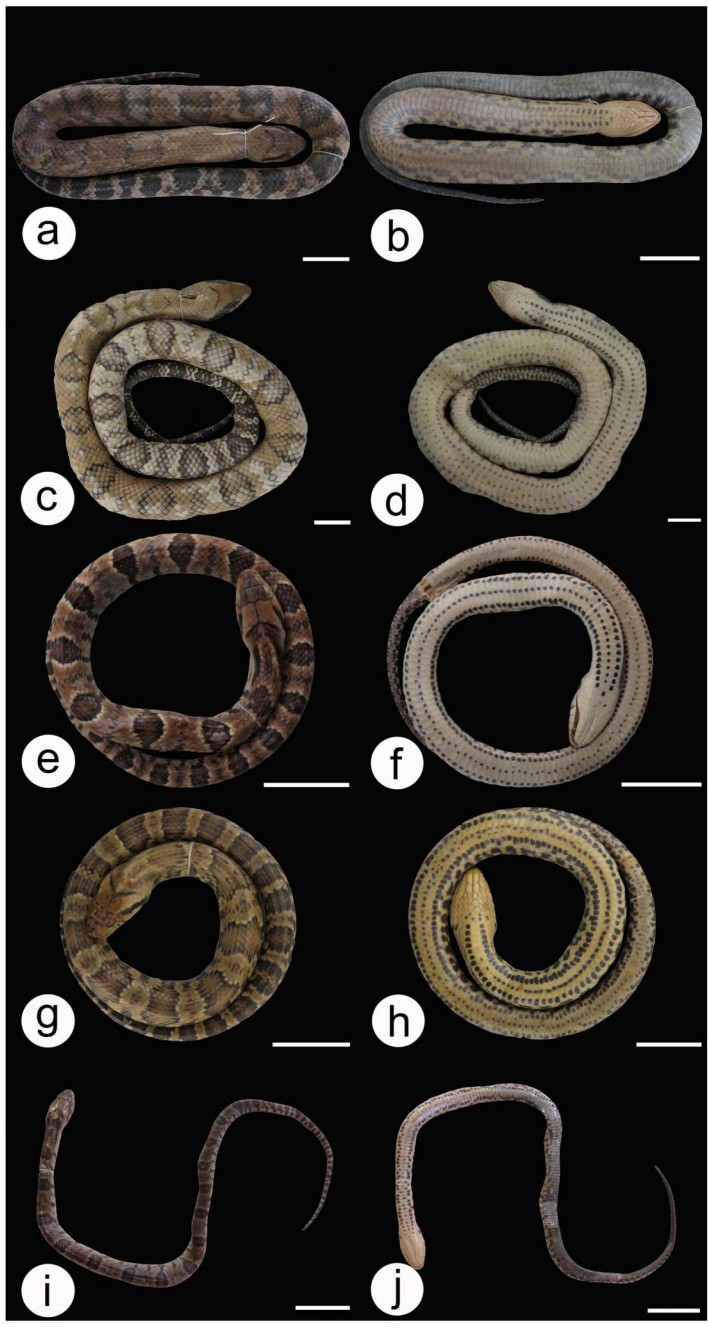

**Figure 7 Dorsal and ventral view of the ontogenetic variation in *Hydrodynastes gigas*.** Amazon basin CHUNB 66534 (A, B); Parnaiba basin CZDP 077 (C, D); East Brazil CHUFPB 14837 (E, F); La Plata basin ZUFMS 1603 (G, H); Tocantins-Araguaia basin CHUNB 22053 (I, J). Scale of 30 mm.

*& Malcolm, 1994*; *Pellegrino et al., 2005*; *Rocha et al., 2015*), this clearly is not the case for the genus *Hydrodynastes* (see *Murta-Fonseca, Franco & Fernandes, 2015*).

Here, we used an integrative taxonomic approach and adopted the species concept of one lineage with distinct evolutionary histories (*De Queiroz, 2007*), to test the taxonomic validity

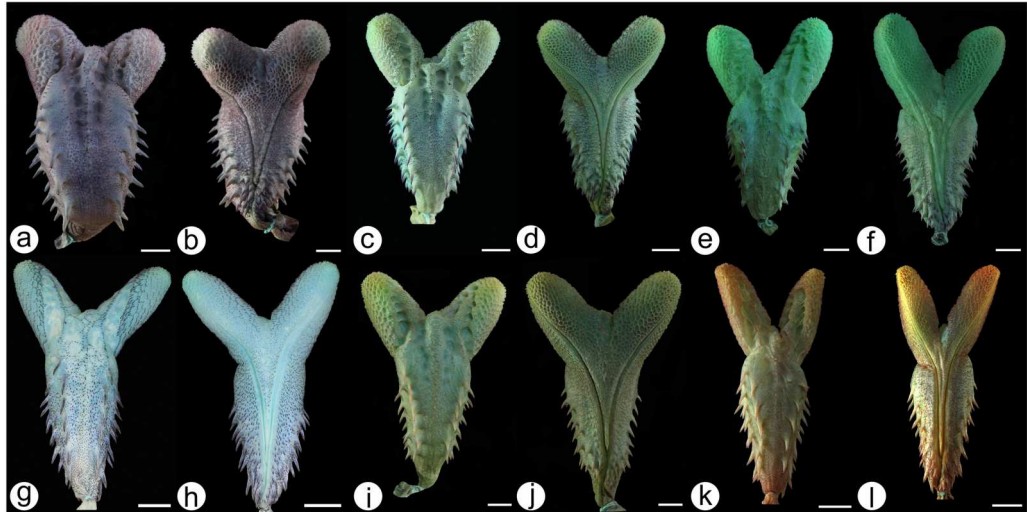

**Figure 8   Hemipenial morphology.** *Hydrodynastes gigas*: Amazon basin MZUSP 18572 (A) asulcate and (B) sulcate side. Northeast South America basin MPEG 25438 (C) asulcate and (D) sulcate side. La plata basin ZUFMS 1910 (E) asulcate and (F) sulcate side; MHNCI 4511 (G) asulcate and (H) sulcate side; UFSM 1937 (I) asulcate and (J) sulcate side. *Hydrodynastes melanogigas* Tocantins-Araguaia basin CHUNB 12802 (K) asulcate and (L) sulcate side. Scale of 10 mm.

of *Hydrodynastes melanogigas*. Our results did not separate *H. gigas* and *H. melanogigas* based on molecular, meristic, morphometric and hemipenial characters. The hemipenis of *Hydrodynastes melanogigas* analyzed showed no differences from the hemipenis of *H. gigas* (*n* = 19). In their description of *H. melanogigas*, *Franco, Fernandes & Bentim (2007)* also pointed out its similarity with *H. gigas* based on meristic characters and hemipenis morphology. The only superficial distinction that remains between these two taxa is the presence of melanism in the latter. Geographic or regional melanism has already been reported for several populations of Squamata (*Pearse & Pogson, 2000*; *King, 2003*; *Bernardo et al., 2012*). In addition, polychromatism can be a bias within taxonomy, if the revision and/or description of species does not take into account the organisms throughout their whole distribution (*Bernardo et al., 2012*; *Ruane et al., 2018*; *Mângia et al., 2020*). In fact, *Franco, Fernandes & Bentim (2007)* carried out an analysis covering almost the entire distribution of *H. gigas*, however we found degrees of melanism in some populations that were not identified by other authors. The variation of melanism found in *H. gigas* and the melanistic gradient observed in *H. melanogigas* (Figs. 4 and 5), along with genetic support, suggests that *H. melanogigas* is not a distinct species but rather a melanic population of *H. gigas*. The distribution of *Hydrodynastes melanogigas* without sympatry with *H. gigas* in the Tocantins-Araguaia basin was an important factor for its description (*Franco, Fernandes & Bentim, 2007*). However, in this study, we analyzed two juveniles of *H. gigas* from the type locality of *H. melanogigas* (CHUNB 22053, Figs. 7I–Figs. 7J; CHUNB 22068). All specimens analyzed by *Franco, Fernandes & Bentim (2007)* and herein were adults or juveniles and no neonates were observed. Therefore, we do not know whether the specimens considered as *H. melanogigas* could have been born melanic or if melanism occurs during their

ontogeny. Still, some studies suggest that thermal melanism is associated with latitude and high altitudes, i.e., relatively cold environments (*Capula, Luiselli & Monney, 1995*), which does not agree with the present case. More studies are needed to confirm the adaptive meaning of melanism through studies of thermal biology. Therefore, due to the lack of any characteristic that sustain these two taxa as distinct species and their low genetic distance (0.04% 16S and 0.2% Cytb), we consider *H. melanogigas* Franco, Fernandez & Bentim, 2007 as a junior synonym of *H. gigas* (*Duméril, Bibron & Duméril, 1854*).

## Systematic account

*Hydrodynastes gigas* (*Duméril, Bibron & Duméril, 1854*)
*Xenodon gigas Duméril, Bibron & Duméril, 1854*. Erpétologie générale vol. 7: 761.
 *Lejosophis gigas Jan, 1863*. Elenco Sistematico degli Ofidi descritti e disegnati per l'Iconografia Generale. vol. 2: 56.
*Cyclagras gigas Cope, 1885*. Proceeding of the American Philosophical Society: 185.
*Cyclagras gigas Boulenger, 1894*. Catalogue of the snakes in the British Museum vol. 2: 144.
*Lejosophis gigas Dunn, 1944*. Caldasia: 69.
*Lejosophis gigas Hoge, 1958*. Papéis Avulsos de Zoologia: 222.
*Hydrodynastes gigas Hoge, 1966*. Ciência e Cultura: 143.
 *Cyclagras gigas Peters & Orejas-Miranda, 1970*. Catalogue of the Neotropical Squamata. Part I: 78.
*Hydrodynastes gigas Dowling & Gibson, 1970*. Herpetological Review (2): 38
*Hydrodynastes melanogigas Franco, Fernandes & Bentim, 2007* Zootaxa (1613): 58. **New Synonymy**

Type material: syntype MNHN 3623
Type locality: Corrientes Province, Argentina.
   **Comments on the type series:** Duméril, Bibron and Duméril's (1854) description of *Xenodon gigas* did not refer to any voucher specimen. The authors only cited that M. A. d'Orbigny collected three individuals in Rio de La Plata, Corrientes Province, Argentina (without further information). The authors also mentioned a plate (Xénodon géant. Atlas, pi. 76, fig-5), which only presents a cranial picture. Overall, *Wallach, Williams & Boundy (2014*, p. 339) indicate that the types would be MNHN 2493 a-c, but according to *Uetz et al. (2019)*, it would be an individual labeled MNHN 3623. Due to this conflicting information, we contacted the curator of the Herpetological Collection at the Muséum National d'Histoire Naturelle de Paris (Dr. N. Vidal) who confirmed that there is only one type specimen, a skin labeled MNHN 3623 (Fig. 9), deposited in the collection, and that the other two specimens appear to be lost . Since *Duméril, Bibron & Duméril (1854)* did not select any specific specimen from their type series, we therefore, designate the specimen MNHN 3623 as the lectotype of *Hydrodynastes gigas*.
   **Description of the lectotype MNHN 3623** (Fig. 9): Adult of undetermined sex; SVL 1570 mm; TL 540 mm; HL 62 mm; HW 35 mm; DN 10 mm; two internasals; nasal divided;

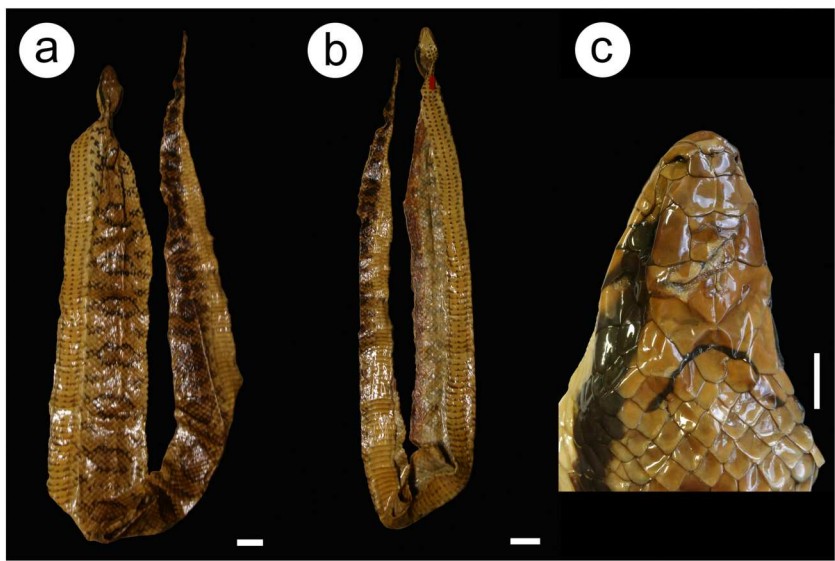

**Figure 9** **Lectotype MNHN 3623 Hydrodynastes gigas.** Dorsal (A) and ventral (B) views of the body and head view (C). Credit: Hussam Zaher. Scale of 50 mm for body view and scale of 10 mm head view.

one loreal; one preocular; three suboculars; two postoculars; temporal $2+2/2+2$; nine supralabials, none contacting the orbit; eleven infralabials, first to sixth contacting chin shields on the right side and first to fifth on the left side; two pairs of chin shields; dorsal 19/16/15 scales, smooth; two apical pits; ventral 153; subcaudals 74, paired and cloacal scale single.

**Color of the preserved lectotype (ethanol 70%)** (Fig. 9): Head brown with black 'U' shaped spot at the end of the parietal scale; post-ocular stripe that extends longitudinally on each side; supralabial brown with the last four scales stained black; infralabials and chin shields cream; dorsum of body brown with dark rounded spots that extend to the end of the tail; ventral body cream with three black stripes to the middle of the body.

**Diagnosis:** *Hydrodynastes gigas* is distinguished from its congener *H. bicinctus* by the following combination of characters: dorsal scales normally 19/19/15; ventral scales in males 150–168 and in females 152–172; subcaudal scales in males 58–88 and in females 49–84; maxillary teeth 15–17; two apical pits in the dorsal scales; post-ocular stripe that extends longitudinally (on each side); ventral body with three lines of continuous spots up to the middle of the body.

**Variation**: All variation in morphometric and meristic data are presented in Tables 3 and 4. Regarding coloration patterns, a considerable degree of color variation can be found in *H. gigas* (Figs. 4A–4L; 5A–5L; 6A–6R and 7A–7J) dorsum ranging from yellow to dark brown or completely dark in melanic populations; rounded spots on the dorsum may vary in shape and size, in some individuals they may be hollow or filled; in neonates the dark rounded spots are well defined throughout the dorsum until the end of the tail, and these spots are outlined by a lighter line; darks spot in the shape of 'V' or 'U' at the end of the parietal scale, only visible in non-melanic populations; ventral body cream/brown with

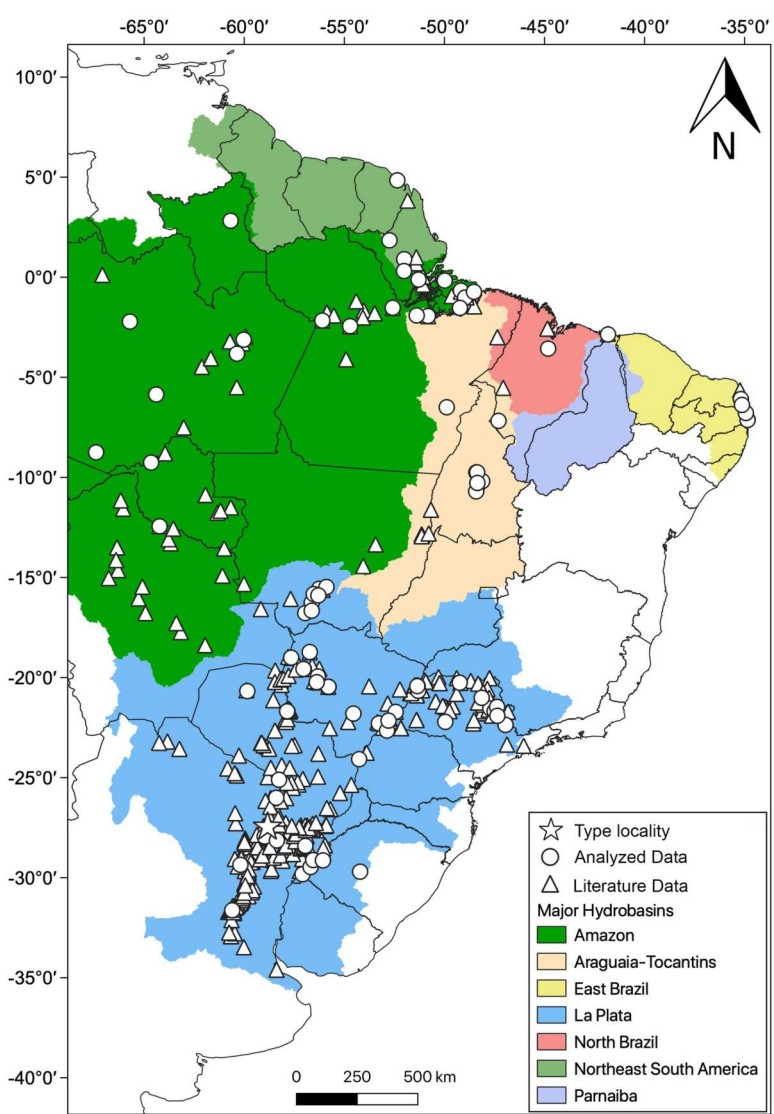

**Figure 10** **Geographic distribution of *Hydrodynastes gigas* in South America.** We compiled data from the literature, specimens and tissues analyzed.

three black stripes that usually go up to the middle of the body, rarely to the cloaca, some individuals do not have these stripes, in melanic populations the belly is dark without spots or when present these spots are continuous on the sides.

**Distribution**: *Hydrodynastes gigas* is widely distributed throughout South America, east of the Andes, occurring in Amazon, East Brazil, La Plata, North Brazil, Northeast South America, Parnaiba and Tocantins-Araguaia hydrobasins (Fig. 10).

## CONCLUSIONS

Our results did not separate *H. gigas* and *H. melanogigas* based on molecular, meristic, morphometric and hemipenial characters. Therefore, the melanistic pattern

of *Hydrodynastes melanogigas* is characterized here as the result of polymorphism within *H. gigas*. Although our integrative approach helped elucidate the taxonomic status of *H. melanogigas*, we believe future, multi-loci phylogeographic studies are needed in order to better understand the evolutionary history of the populations belonging to the two remaining species *H. gigas* and *H. bicinctus*.

## ACKNOWLEDGEMENTS

We are grateful to G. Puorto and F. Grazziotin (IBSP), G. Colli (CHUNB), D. Mesquita and G. Vieira (CHUFPB), J. Lima (IEPA), P. Manzani (ZUEC), G. Graccioli (ZUFMS), S. Cechin (ZUFSM), C. Previero (MZCEULP), A. Guzzi (CZDP), F. Resende (FUNED), R. Oliveira (MCN), G. Pontes (MCP), JC. Leite (MNHCI), A. Prudente (MPEG), F. Curcio (UFMT), P. Passos (MNRJ), and N. Vidal (MNHN) for allowing access or loaning specimens under their care. We are indebted to Ana Bottallo Quadros for verifying and photographing the type specimen at the MNHN. We also thank Diego Cavalheri and Roberta Murta-Fonseca for verifying and taking photos from *H. melanogigas* type series specimens at the IBSP and MNRJ.

### Funding

This work was supported by Coordenação de Aperfeiçoamento de Pessoal de Nível Superior (CAPES Finance Code 001), Universidade Federal de Mato Grosso do Sul – UFMS/MEC – Brasil, CNPq (Conselho Nacional de Desenvolvimento Científico e Tecnológico) provided research fellowships (311492/2017-7 and 309950/2018-0) and this research was supported by grant 2011/50206-9 to Hussam Zaher. There was no additional external funding received for this study. The funders had no role in study design, data collection and analysis, decision to publish, or preparation of the manuscript.

### Grant Disclosures

The following grant information was disclosed by the authors:
Coordenação de Aperfeiçoamento de Pessoal de Nível Superior.
CNPq (Conselho Nacional de Desenvolvimento Científico e Tecnológico): 311492/2017-7, 309950/2018-0.

### Competing Interests

The authors declare there are no competing interests.

### Author Contributions

- Priscila S. Carvalho conceived and designed the experiments, performed the experiments, analyzed the data, prepared figures and/or tables, authored or reviewed drafts of the paper, and approved the final draft.
- Hussam Zaher and Nelson J. da Silva Jr conceived and designed the experiments, authored or reviewed drafts of the paper, and approved the final draft.
- Diego J. Santana conceived and designed the experiments, analyzed the data, prepared figures and/or tables, authored or reviewed drafts of the paper, and approved the final draft.

## DNA Deposition

The following information was supplied regarding the deposition of DNA sequences:

The 16S sequences are available at Genbank: MT192267–MT192299; MT215327–MT215335. The Cytb sequences are available at Genbank: MT224972–MT225014.

The Cmos sequences are available at Genbank: MT328065–MT328096.

The NT3 sequences are available at Genbank: MT328097–MT328134 and MT424769. In addition, all information is available in Table 1.

## Data Availability

Summarized morphometric and meristic data are available in Tables 3 and 4. Location data and zoological collections are available in Appendix S1. The data shows overlap in all meristic and morphometric characteristics between *Hydrodynastes gigas* and *H. melanogigas*.

## Supplemental Information

Supplemental information for this article can be found online at http://dx.doi.org/10.7717/peerj.10073#supplemental-information.

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
