# Peer review of "A morphological and molecular study of Hydrodynastes gigas (Serpentes, Dipsadidae), a widespread species from South America"

_PeerJ, doi:10.7717/peerj.10073_

## Round 0.1 · original submission · Major Revisions

Thank you for submitting your work to PeerJ. I have sent your paper to expert referees for their consideration. I have now received their comments back and have read through your paper carefully myself. Enclosed please find the reviews of your manuscript.

The reviewers find some merit in your work, however, they raise a number of important questions and concerns which have to be addressed prior to acceptance of your paper to publication. Some of these issues are critical for interpreting your data and taxonomic results, for example, the request of Reviewer 2 to provide PCA or discriminant analysis of your morphological dataset.

Therefore I would ask you to revise your manuscript in accordance with the suggestions of the reviewers. After I get the revised version of your paper it will be forwarded to the same referees for consideration. In your rebuttal letter please address all questions raised by the reviewers on point by point basis.

Reviewer 1 ·

Basic reporting

The current study is well addressed and showed important results for the scientific community. However, some points need to be clarify for a better understanding, especially on the discussion section. The discussion needs to better addressed according to the goals of the study. I also suggest the authors to incorporate a strong relationship about the weak and strong points of their results with a better interpretation.

Below I point out some my suggestions to improve the manuscript:

Title:
The title of the paper mentions the different hydrographic basins from South America, however, I do not see along the text any discussion about the different basins. What are these basins? How they differ from each other and how it affects your results? It will be also good to see the different basins with the rivers on a figure 1.

Abstract:
I suggest that you improve the statement at lines 27-29, I do not understand which populations are you mention here if before you were talking about the species.

Text:

For the introduction, I would like to see the problematic of the species group for the reader to be able to follow the context.

Lines 71-72: When you explain about the distribution of Hydrodynastes gigas in Brazil, I suggest you to mention the Brazilian states as you did for H. melanogigas.

Lines 75-77: When you mention about “… the morphological and molecular variations between its populations”, I also suggest you to mention between population and species. Hydrodynastes is a widely distributed genus with different species as you mention before.

Lines 77-78: I suggest you to explain more about the influence of the basins on the speciation of aquatic and semiaquatic fauna. This sentence is out of the context and I did not see any explanation before in your text about speciation process or ecological differences along species and populations.

Lines 79-80: I suggest you to mention the two species name that you are going to evaluate the taxonomic status.

Line 80: Change “Thus, our objectives were” for “Thus, our objectives are”

Lines 80-82: I suggest you to provide a better justification for your goals here. Are you going to test whether Hydrodynastes gigas comprises a complex of cryptic species based on? Also it will be better addressed if you can explain a little more how you are going to verify the taxonomic validity of H. melanogigas.

Lines 87-89: I suggest the authors to provide a justification for the use of 3 species, if at the introduction (Lines 80-82) they only talk about 2 species.

Line 95: A reference is missing “in the Geneious v.9.1.8 program”.

For the methods section I suggest the authors to provide a better explanation about the use of those methods according to their goal and hypotheses.

Line 129: I suggest you to subset the methods for “Hemipenis morphology” as you did for the discussion.

Line 130: Change “Amazon” for “Amazonia”, also on all the text.

Line 181: I suggest the authors to start the discussion with a general statement about their main results and conclusion, then start to discuss about this.

Lines 183-186: I don’t see any link about the influence of rivers on the speciation process along your text. It seems very out of context here. Any time along your introduction and methods you addressed biogeographic methods or hypotheses, so I don’t think it is well included here. Also the references are only about terrestrial organisms, so you cannot talk about aquatic species here.

Lines 211-213: the reference is missing.

Line 215: it is missing the reference about the “other authors”

Line 304: Change “…besides to elucidating the events of the diversification of aquatic and semi-aquatic species and Squamata.” for “…besides to elucidating the diversification events of aquatic and semi-aquatic Squamata species.”

--

Figure 1: It is really hard to see the different shades of gray in the map (the basins). I suggest you to change it. Also I suggest you to improve the legend of the map. When you say that the map B “specimens analyzed” , what kind of data are you mention? Please explain better here.

Figure 2: During your Results section you talk about the sample of H. melanogigas in the Figure 2. However, it is not clear in the figure which one are these samples. Also you need to provide a better statement about the Figure, for example the samples marked with starts are not clear on the legend.

Table 1: Change “Table 1. Specimens used in the molecular analyses,..” for “Table 1. Specimens used for the molecular analyses,..”.

Table 2: I suggest the authors to improve the legend giving more details about the models.

Table 3: Please include a statement about the bold number on the legend.

Experimental design

I commend the authors for their extensive data set. The methods are good, but if there is a weakness I would say that a strong relationship and a better explanation about the method with the hypothesis that the authors are testing would improve the manuscript. Further, most of the figures’ legends need more details or/and are missing something, I suggest to improve them.

Validity of the findings

I think that the findings of the study are very important for science. As I mention before the date is robust and well analyzed. My suggestions made above will improve the manuscript and also impacted the conclusion, which support the results.

Additional comments

In general the paper is good and shows important results for the scientific community. I am glad to see this work and I hope the changes are going to improve the paper. Some points need to be clarify for a better understanding, especially on the discussion section.

Finally, I think that is very important for any study to mention the Funding, which are missing in the current project.

Reviewer 2 ·

Basic reporting

The study by Carvalho and collaborators compile data on genetic, morphological, and coloration patterns of Hydrodynastes species in order to identify the taxonomic status of the species in the genus. The research topic and study are interesting, as I am familiar with the doubt related to the validation of H. melanogigas status as a species. However, there are a few issues that require some attention from the authors. The Introduction needs work, improving the connection of ideas, and highlighting the importance of the study (contextualization). For example, you mention about variation on color patterns, but don’t discuss why it is relevant for the species in question. More context is needed for the relation of hydrographic basins and genetic structure expected, highlighted in the figures and the title, but briefly mentioned in the discussion and not included in the Introduction. This restructuring of the Introduction will need to be followed in the Discussion. The English language needs an overall improvement to ensure a clear understanding of the text. More details are mentioned below to help the authors in revising the manuscript.

Experimental design

If the changes in the Introduction improve the sets up of the arguments of the study, the experimental design will make more sense, as I recognized the question as meaningful. Then, as the genus Hydrodynastes only includes three species and genetic data on H. bicinctus is included, I missed the addition of morphological, coloration patterns, and hemipenis morphology description for the species. It would make an interesting parallel on the conserved morphology founded on H. gigas/melanogigas. I would also suggest authors consider running an additional analysis (PCA or discriminant analysis) on the morphological data for easy visualization of the morphological variation in both species.

Validity of the findings

While I recognized some changes that would improve the quality of the manuscript,
the taxonomic conclusions are supported as they can be, given the lack of genetic and morphological variation. As only one hemipenis of H. melanogigas was analyzed, a comparison to published data would improve the support of the statements made.
Then, while the genetic tree does not support H. melanogigas as a distinct species, it does appear as a separate group, and as a relatively geographically isolated population of H. gigas. However, I am fine to let future studies and the community to decide whether to accept the taxonomic decision.

Additional comments

Abstract – background: please specify what integrative approaches means
Line 24: Hydrodynastes melanogigas in italic
Line 38 to 46: The paragraph is confusing and needs some rewording.
Line 48: “database” is a weird word for this statement
Line 54: please cite more studies
Line 79: I missed more details on the aims of the study. A paragraph explaining why a taxonomic study of the group is important, why you would expect H. gigas to be a complex of cryptic species, the relation with hydrographic basins. You mention briefly on line 75 but more details are needed, and references should be included to support.
Line 78: Please include a reference to your statement
Line 87: remove the word “total”
Line 203-204: more hemipenis morphology data for H. melanogigas, or comparison to other studies would reinforce your statement, as you only have the hemipenis of one individual
Line 302: highly recommended sounds weird
Line 304: broad; please be more specific
References: Please check reference format of the journal e.g. full title of the journal should be in italic
Figure 1: “a” and “b” should be in caps lock; please remove the last zero of the coordinates (e.g. 10 instead of 100); be consistent on the symbol used for the species on the maps a and b
Figure 2: specify what the asterisk means in the legend
Figure 3 to 6: can you please specify which ones were H. melanogigas? Also, avoid acronym on the legend, writing Hydrodynastes instead of just “H. “
Table 1: consistency - et al in italic or not; can you check the localities of the samples with the authors of the papers?
Table 3: appears in the text after table 4 and 5; on the table, please substitute the numbers 1-5 for the name of species, for clarity

---

## Round 0.2 · Minor Revisions

Thank you for submitting your manuscript to PeerJ. I have sent your paper to expert referees for their consideration. I have now received their comments back and have read through your paper carefully myself. Enclosed please find the reviews of your manuscript.

The reviews are in general favourable and suggest that, subject to minor revisions, your paper could be suitable for publication. Please consider these suggestions, and I look forward to receiving your revision.

Reviewer 1 ·

Basic reporting

Below I left some of the improvements that need to be addressed, however the paper needs a minor revision.

Text:

Lines 17-178: The sentence does not show the results. You examined fully melanic specimens and I would like to know what you found here.

Lines 186-189: I didn’t understand this sentence at the results section. It should be at the methods section and I didn’t see below the “variation” that you want to mention. I suggest you to change it.

Line 210: I suggest you to remove the word “however” and put a comma. The way these sentences are structure now don’t mean what you want to say.

Lines 219-221: I would like to see a study example (reference) about this statement.

Figure 1: At the title of the Figure 1A you mention only Hydrodynastes but at the legend you mention the species names. Just be consistent with the names. I also suggest you to combine the maps so we can see the whole distribution of the species.

For most of the figures, be consistent with the letter style, for examples in the text the Figures letters are uppercase (Fig. 4F) and in the figures they are lowercase (4f). I don’t know if is the journal style.

Figure 10: I suggest you to combine the figure 1 with the figure 10 and explain in the legend the different samples used for the analyses and literature (maybe different colors).

Experimental design

I am glad to see the improvement on the manuscript from the last time. The authors made a good job accepting the suggestions. In general, the text has a better flow now and is easier to follow.

Validity of the findings

I am glad to see the improvement on the manuscript from the last time. The authors made a good job accepting the suggestions. In general, the text has a better flow now and is easier to follow.

Additional comments

I am glad to see the improvement on the manuscript from the last time. The authors made a good job accepting the suggestions. In general, the text has a better flow now and is easier to follow. However, some parts are still need improvement for the publication. At the discussion section, the structure needs to follow the rest of the paper, for example if you talk first about molecular, morphological and hemipenis data on the other sections, you need to do the same at the discussion. Also, I suggest you to integrate more the paragraphs for the discussion, try to see the big picture here, combining the all the results that you have instead of discussing them separated. The third paragraph (Line 222) is a good example of integration, but the fist 2 paragraphs are a bit disconnected. I accepted the paper with minor revision.

Reviewer 2 ·

Basic reporting

I think the revised version is much improved and almost ready for publication- there are still some points needing clarification that I would like to mention. Most of my comments are suggestions, that can or cannot be adopted by the authors unless the editor believes the changes are necessary for publication. The only bit that I believe really needs to be solved is if the authors removed the allometric effect of the body size for the PCA analysis (see more below).

The title and abstract are good, but in my opinion, the first paragraph of the Introduction needs some rewording. The idea is there, but the repetitiveness of the “however, however, on the other hand, thus” seems a bit odd. Here is just an idea to try to help, rewording your words, but of course (please) feel free to edit as you want – “Taxonomic studies on species description are traditionally based on morphological characters to delimit species. However, in many cases, species are difficult to delimit due to the limited number of morphological differences or the absence of them, preventing the recognition of valid cryptic species. Morphology alone might result in more than one name being assigned to individuals belonging to the same evolutionary lineage (i.e. species). This result in many species been described based solely on morphological parameters, which could merely reflect interpopulation variation, instead of evidence of lineage separation”.

On the methods, line 107-108 you mention “using the standard Polymerase Chain
Reaction (PCR) technique”. The ideal is that you mention in the Supplementary Material the time and temperature in each cycle of the PCR, or the paper that shows it, to facilitate replication of your study. For the PCA analysis, have you removed the allometric effect of the body size? If yes, please mention in the text. If not, this is a necessary step, as the size is associated with individual growth, and you are focused on the shape (that must be size-free). Please refer to many studies and different normalization methods.

On the discussion, line 210, please remove “. However,”. For consistency with the way the authors presented the methods e results, I would move the paragraph about hemipenis (line 212-221) to line 245, before the last paragraph (“More studies are needed”).

I would move Figure 10, maybe Table 1, and Table 2 to the Supplementary Material.

Experimental design

No comment

Validity of the findings

No comments

Additional comments

I hope the suggested changes are going to improve the paper. Great work.

---

## Round 0.3 · accepted · Accept

Thank you for taking the time to revise and resubmit your manuscript. I have now read through your paper as well as your letter in response to the reviews. I think that you have successfully addressed all of the concerns raised very well, and would like to accept your manuscript for publication in PeerJ. Congratulations.

Thank you for all the hard work you have put in to this. Your paper makes a strong contribution to the literature and I look forward to seeing it published.